



# Centroids in second-order conservative remapping schemes on spherical coordinates

Fuyuki SAITO[1]

[1]Japan Agency for Marine-Earth Science and Technology (JAMSTEC), Yokohama, Japan

**Correspondence:** Fuyuki SAITO (saitofuyuki@jamstec.go.jp)

**Abstract.** The transformation of data from one grid system to another is common in climate studies. Among the many schemes used for such transformations is second-order conservative remapping. In particular, a second-order conservative remapping scheme to work on the general grids of a sphere, either directly or indirectly, has served as an important base in a variety of studies.

In this study, the author describes a fundamental problem in the derivation of the method proposed by a pioneer study relating to the treatment of the centroid used as a reference point for the second-order terms in the longitudinal direction. In principle, use of the original formulation has a potential to cause damage to the entire remapping result. However, a preprocessing procedure on the longitude coordinate suggested in the algorithm for other objectives tends to minimize or even erase the error as a side effect in many, if not most, typical applications. In this study, two alternative formulations are proposed and tested and are shown to work in a simple application.

## 1 Introduction

Numerical climate models commonly couple individual component models such as models for atmosphere, ocean, and land. These component models are typically developed as stand-alone models and often adopt their own grid system for efficiency. Coupling between such components involves field transformations of data from one grid system to another, while preserving key attributes of interest, e.g., global and/or local integrals. This procedure for conservative quantities is often referred to as *conservative remapping*(e.g. Dukowicz and Kodis, 1987). As summarized in Mahadevan et al. (2022), there have been considerable efforts to create conservative remapping algorithms for various problems.

Remapping algorithms used in global climate studies are typically based on first- and second-order conservative mesh-based schemes (Mahadevan et al., 2022). In the first-order conservative scheme, a conservative quantity assuming a constant distribution over the source grid cell is transformed into the overlapped destination grid cells with area-weighted remapping (Bryan et al., 1996). On the other hand, in the second-order conservative scheme, a linear distribution within a source grid cell is assumed, which results in a more accurate and smoother transformation than is the case for first-order schemes. In particular, a second-order algorithm works efficiently when remapping from spatially coarse resolution to fine resolution. Because of this, it is considered the preferred choice in many remapping applications. Dukowicz and Kodis (1987) (hereafter referred to as DK87) first provided a second-order conservative remapping algorithm that works for any general grid system using





Gauss's divergence theorem for simplification of area integrals converted into line integrals. According to Taylor (2024), most conservative remapping algorithms are variants of this approach (there is also a good summary of the remapping method in the appendix of Taylor, 2024).

Jones (1999), hereafter referred to as J99, extends the DK87 theory to spherical coordinates, offering an approach that can be applied to any type of grid on a sphere. Many efforts to maximize efficiency are included in the proposed algorithm, and a number of problems essentially originating from the spherical coordinate system are solved. J99 also provides the Spherical Coordinate Remapping and Interpolation Package (SCRIP), a native software to implement the algorithm (see https://github.com/SCRIP-Project) in addition to four other remapping methods. SCRIP is one of the most widely used remapping software packages in the climate community (Ullrich et al., 2009). For example, Climate Data Operators (CDO) (Schulzweida, 2023) have once included a conservative remapping option that incorporates SCRIP with rewriting the source code from Fortran to ANSI/C (which is excluded from the latest version, though). In addition, SCRIP has been adopted by the general coupler library OASIS3-MCT_3.0 (Craig et al., 2017), which is used by many modeling groups.

The algorithms and the software proposed in J99 have, either directly or indirectly, been an important base in a variety of studies, including both observational and model data analyses(e.g. Barnes et al., 2024), as well as numerical model development (e.g. Ding et al., 2024). Recently, some softwares use implementations for the second-order conservative remapping which significantly differ from the J99 algorithm (e.g. Kritsikis et al., 2017) and the communities have been switching to the other software. However, as far as the author surveyed, some recent studies still use the J99 scheme for the second-order conservative method, as explicitly mentioned in, e.g., Ding et al. (2024), Chtirkova et al. (2024) and Ren and Zhou (2024).

Despite this widespread acceptance, however, there appears to be one distinct and fundamental problem in the derivation of core equations in J99 (Eq. 10) that, to the author's knowledge, has not previously been recognized nor reported.

The problem is in the treatment of a reference point to evaluate the second-order term in the longitudinal direction. In J99, one of the core equations is, at the very end, transformed into an invalid formulation. If one implements the J99 algorithm following the equations, in particular Eq. (10), as presented, there is a risk that serious damage will be caused to the remapping result.

Although few, if any, studies using the second-order conservative remapping scheme in SCRIP have reported strange or erroneous behavior, this is not because the derivation is valid. Rather, there is a small preprocessing block in the algorithm suggested in J99, that adjusts some of the key variables for possibly other objectives which can mask the fundamental problem as a side effect. With this adjustment, any errors originating from the invalid derivation tend to be minimized. In fact, the errors can be fully canceled when the source grid cell is a simple one, such as a regular latitude-longitude (RLL) rectangle grid.

In the next section, the basics of second-order conservative remapping methods are described, with a proposal for a consistent formulation of the scheme. The fundamental problem in the J99 derivation is identified, and the reasons why the invalid derivation has not heretofore been revealed as a problem are discussed. In the third section of the paper, the influence of the inconsistent formulation is demonstrated in simple but practical cases. An experiment showing a sample implementation of the proposed schemes is presented.



## 2 Description of the second-order remapping methods

This section describes the basic idea of the second-order conservative remapping scheme of DK87 and its extension to the spherical coordinate system as formulated by J99, with supplementary explanation by Jones (2024). The original equations and terms are transformed into the formulation shown in J99. For example, the volume integral notation in DK87 is replaced by the surface integral in accordance with J99. Additionally, some new symbols unique to the present paper are introduced for

description.

### 2.1 Derivation on a general case

Below is the set of equations derived in a slightly different way, partially following the DK87 method. The derivation is somewhat roundabout but is necessary for the flow of the present paper. Some trivial descriptions are included so as to avoid ambiguities, which the author believes is necessary in the context of the present paper.

The object is to compute in a conservative manner, a flux term on a destination grid from the flux term on a source grid over a surface of three-dimensional Euclidean space. For any flux terms that must satisfy a constraint to preserve conservation, the flux integral over each source grid cell must be consistent with the average value in the grid cell as follows:

$$\overline{f}_n A_n = \int_{A_n} f_n \, \mathrm{d}A, \tag{1}$$

where $n$ is the source cell index, $f_n$ and $\overline{f}_n$ are a flux term and its average over the area of source cell $n$, respectively.

Equation (1) corresponds to Eq. (19) in DK87. Also, Eq. (1) is identical to Eq. (4) in J99. Hereafter, Eq. ($e$) in J99 are referred to as Eq. (J99.$e$) in order to avoid confusion.

DK87 proposes to approximate the source flux by a combination of the average and its gradient. Assuming the flux gradient is constant across a source grid cell locally, the flux can be approximated around a reference point as follows:

$$f_n = f(\mathbf{c}_n) + \nabla_n f \cdot (\mathbf{r} - \mathbf{c}_n), \tag{2}$$

where $\mathbf{r}$ is the position vector, $\mathbf{c}_n$ is the position vector of a reference point (corresponding to $\overline{\mathbf{r}}_k$ in DK87, Eq. 20) and $\nabla_n f$ is a gradient of $f$ in source grid cell $n$. As mentioned in DK87, Eq. (2) is equivalent to the first two terms in the Taylor series expansion of $f$ about the point $\mathbf{c}_n$. However, the formulation is actually chosen independently of the Taylor expansion among a number of possible solutions which satisfies the conservation characteristics (Jones, 2024). The reference $\mathbf{c}_n$ can be chosen arbitrarily in the source cell; here, it is defined such that the flux approximation Eq. (2) satisfies the condition Eq. (1). By

substituting $f_n$, the following condition is obtained:

$$\overline{f}_n A_n = f(\mathbf{c}_n) A_n + \int_{A_n} \nabla_n f \cdot (\mathbf{r} - \mathbf{c}_n) \, \mathrm{d}A. \tag{3}$$





In order to satisfy Eq. (3) for any flux and its gradient, the following constraints are obtained:

$$\overline{f}_n = f(\mathbf{c}_n), \tag{4}$$

$$\int_{A_n} \nabla_n f \cdot (\mathbf{r} - \mathbf{c}_n)\, \mathrm{d}A = 0. \tag{5}$$

Given a constant gradient across source grid cell $n$, the gradient terms in Eq. (5) can be taken out of the integral, and the condition is simplified as follows:

$$\int_{A_n} \mathbf{r}\, \mathrm{d}A - \int_{A_n} \mathbf{c}_n\, \mathrm{d}A = 0. \tag{6}$$

Equation (6) is the principle condition of the the reference $\mathbf{c}_n$ term. Under Eq. (6) constraints with Eq. (4), the flux approximation (Eq. 2) automatically satisfies the conservation characteristics of Eq. (1). Conservation is preserved with second-order
accuracy if the gradient is at least a first-order approximation; if the second term is neglected, the method corresponds to the first-order method.

At least over the three-dimensional Cartesian coordinate system, $\mathbf{c}_n$ term can be taken out of the integral also. In this case, the reference $\mathbf{c}_n$ can be inverted as

$$\mathbf{c}_n = \int_{A_n} \mathbf{r}\, \mathrm{d}A \left/ \int_{A_n} \mathrm{d}A \right. = \frac{1}{A_n} \int_{A_n} \mathbf{r}\, \mathrm{d}A, \tag{7}$$

which is identical to the formulation of $\bar{\mathbf{r}}_k$ in DK87, and to $\mathbf{r}_n$ in Eq. (J99.6). The position computed in Eq. (6) or Eq. (7) corresponds to the geometric center, often referred to as the *centroid*, of the source grid cell $n$ under the geometry of the target Euclidean space. In the derivation of DK87, the term centroid is used to label the reference point before the condition in Eq. (6) is presented. While this is viable, the author believes it to be slightly more natural to first describe the condition of the reference point and to subsequently describe its coincidence to the centroid, at least in the context of the present study. Such a derivation
is also given in the next section.

## 2.2 Extension to spherical coordinates

Extension to the spherical coordinates requires to replace the gradient and displacement terms in Eq. (2) in the coordinate system. The gradient term is formulated as follows:

$$\nabla_n f = \left( \frac{\partial f}{\partial \theta} \right)_n \hat{\theta} + \left( \frac{1}{\cos\theta} \frac{\partial f}{\partial \phi} \right)_n \hat{\phi}, \tag{8}$$

where symbols $\theta$ and $\phi$ are adopted for the latitude and longitude coordinates, respectively. The position vector $\mathbf{r} = [x, y, z]^\mathsf{T}$ of the Cartesian coordinate on the unit sphere is expressed using the spherical coordinate components $(\hat{\theta}, \hat{\phi})$ which depend on $\theta$ and $\phi$. The inner product on the spherical coordinate is not simply a component-wise product as in Cartesian coordinates because the direction of the unit vectors depends on the position. J99 approximates that the unit vectors are aligned over the





source cell such that local orthogonality holds true for a simple approach. The local displacement at the coordinate $(\hat{\theta}, \hat{\phi})$ on
the unit sphere can be formulated as

$$\mathrm{d}\mathbf{r} = \hat{\theta}\mathrm{d}\theta + \hat{\phi}\cos\theta\mathrm{d}\phi. \tag{9}$$

Introducing Eq. (9) at the position $\mathbf{c}_n$ and the gradient term Eq. (8) into Eq. (2), the flux term can be approximated as:

$$f_n = \overline{f}_n + \left(\frac{\partial f}{\partial \theta}\right)_n (\theta - \theta_c) + \left(\frac{1}{\cos\theta}\frac{\partial f}{\partial \phi}\right)_n \cos\theta\,(\phi - \phi_c)\,, \tag{10}$$

where the coordinates $\mathbf{c}_n = [\theta_c, \phi_c]^\mathsf{T}$.

The formulation of J99 is mainly derived for the area-averaged flux over the destination grid cell (after remapping), and is
essentially the same as that over the source grid cell. The flux over the destination grid is formulated as follows:

$$\overline{F}_k = \frac{1}{A_k}\sum_{n=1}^{N}\int_{A_{nk}} f_n\,\mathrm{d}A, \tag{11}$$

where $\overline{F}_k$ is the average flux over the destination grid cell $k$, and $A_{nk}$ is the area of the source grid cell $n$ covered by the
destination grid cell $k$. The summation is performed for all overlapped cells of $N$. The average flux term at the destination grid
cell can be approximated with using the flux approximation of Eq. (10), as follows:

$$\overline{F}_k = \sum_{n=1}^{N}\left[\overline{f}_n w_{1nk} + \left(\frac{\partial f}{\partial \theta}\right)_n w_{2nk} + \left(\frac{1}{\cos\theta}\frac{\partial f}{\partial \phi}\right)_n w_{3nk}\right], \tag{12}$$

which corresponds to Eq. (J99.7). The three coefficients, $w_{1nk}, w_{2nk}, w_{3nk}$, are called the remapping weights and are derived
according to J99 as follows:

$$w_{1nk} = \frac{1}{A_k}\int_{A_{nk}}\mathrm{d}A, \tag{13}$$

$$w_{2nk} = \frac{1}{A_k}\int_{A_{nk}}(\theta - \theta_c)\mathrm{d}A, \tag{14}$$

$$w_{3nk} = \frac{1}{A_k}\int_{A_{nk}}\cos\theta(\phi - \phi_c)\mathrm{d}A. \tag{15}$$

The reference point $(\theta_c, \phi_c)$ is actually called the centroid in J99 (expressed as $\theta_n, \phi_n$). Note that Eqs. (14) and (15) are
presented as intermediate formulations (Eqs. J99.9 and J99.10) during the derivation.

Before the final formulations of the remapping weights terms, an important characteristics is described here. It is reasonable
to conclude that Eq. (15) holds for any longitudinal origin; otherwise, the remapping weight $w_{3nk}$ would change its value
according to the coordinate. Thus, in the computation of the weights for each source cell $n$, it would be safe to rotate around
the pole by $\phi_{\mathsf{ofs}}$, which would simply correspond to replacing the longitudinal variable with a relative one. Put formally, Eq. (15)





is reformulated into

$$
w_{3nk} = \frac{1}{A_k} \int_{A_{nk}} \cos\theta[(\phi - \phi_{\mathsf{ofs}}) - (\phi_c - \phi_{\mathsf{ofs}})]\mathrm{d}A
$$

$$
= \frac{1}{A_k} \int_{A_{nk}} \cos\theta[\tilde{\phi} - \tilde{\phi}_c]\mathrm{d}\tilde{A}, \tag{16}
$$

where $\tilde{\phi} = \phi - \phi_{\mathsf{ofs}}$. This is an identity for any $\phi_{\mathsf{ofs}}$. Equation (16) is exactly the same formulation as Eq. (15), thus it can be safely expressed using $\phi$ in terms of $\tilde{\phi}$ without ambiguities. The same replacement from $\phi$ to $\tilde{\phi}$ is performed on the other remapping weights.

J99 suggests to adopt the source grid cell center as $\phi_{\mathsf{ofs}}$ for each source cell instead of the globally-fixed longitude origin. The numerical library SCRIP does include this method. This suggestion is raised from the spherical coordinate system nature, where the longitude is multiple valued on one line on the sphere. Such problems can be easily avoided using this simple method.

Actually, the definition of central longitude is ambiguous for general shapes of the grid cell, which must be supplied by the user according to the source grid cell configuration. Since only the difference between the two relative longitudes adjusted by the offset longitude is used in the computation, the central value is of no particular significance. It is even possible to have the offset longitude fall outside the cell boundaries as far as it is enough to avoid the multiple-value longitude issues. This topic will be discussed later.

The final formulations of $w_{2nk}$ and $w_{3nk}$ conducted in the algorithm are obtained by expanding the reference point $(\theta_c, \phi_c)$. Here, the reference point that corresponds to those defined in J99 is represented as $(\theta_n, \phi_n)$. According to Jones (2024), the position vectors in Eq. (7) are transformed into the corresponding spherical coordinates with including the metric scale factor, as follows:

$$
\theta_n = \frac{1}{A_n} \int_{A_n} \theta \, \mathrm{d}A, \tag{17}
$$

$$
\phi_n \cos\theta = \frac{1}{A_n} \int_{A_n} \phi \cos\theta \, \mathrm{d}A. \tag{18}
$$

Introducing these formulations of the reference coordinate into Eqs. (14) and (15), the final formulations of $w_{2nk}$ and $w_{3nk}$ are as follows:

$$
w_{2nk}^{\mathsf{ORG}} = \frac{1}{A_k} \int_{A_{nk}} \theta \, \mathrm{d}A - \frac{1}{A_k} \int_{A_{nk}} \mathrm{d}A \, \frac{1}{A_n} \int_{A_n} \theta \, \mathrm{d}A = \frac{1}{A_k} \int_{A_{nk}} \theta \, \mathrm{d}A - \frac{w_{1nk}}{A_n} \int_{A_n} \theta \, \mathrm{d}A, \tag{19}
$$

$$
w_{3nk}^{\mathsf{ORG}} = \frac{1}{A_k} \int_{A_{nk}} \phi \cos\theta \, \mathrm{d}A - \frac{1}{A_k} \int_{A_{nk}} \mathrm{d}A \, \frac{1}{A_n} \int_{A_n} \phi \cos\theta \, \mathrm{d}A = \frac{1}{A_k} \int_{A_{nk}} \phi \cos\theta \, \mathrm{d}A - \frac{w_{1nk}}{A_n} \int_{A_n} \phi \cos\theta \, \mathrm{d}A, \tag{20}
$$





which correspond to Eqs. (J99.9) and (J99.10), respectively. As explained above, Eq. (20) is computed using $\tilde{\phi} = \phi - \phi_{\mathsf{ofs}}$ term, the longitude relative to a reference longitude $\phi_{\mathsf{ofs}}$:

$$\tilde{w}_{3nk}^{\mathsf{ORG}} = \frac{1}{A_k} \int_{A_{nk}} \tilde{\phi} \cos\theta \, \mathrm{d}\tilde{A} - \frac{w_{1nk}}{A_n} \int_{A_n} \tilde{\phi} \cos\theta \, \mathrm{d}\tilde{A}. \tag{21}$$

The integral parts of the remapping coefficients $w_{1nk}$, $w_{2nk}$ and $\tilde{w}_{3nk}^{\mathsf{ORG}}$ (Eqs. 13,19 and 21) are computed by transforming it into a line integral using Gauss's divergent theorem (DK87, J99).

$$\int_{A_{nk}} \mathrm{d}\tilde{A} = \oint_{C_{nk}} -\sin\theta \, \mathrm{d}\tilde{\phi}, \tag{22}$$

$$\int_{A_{nk}} \theta \, \mathrm{d}\tilde{A} = \oint_{C_{nk}} -[\cos\theta + \theta\sin\theta] \, \mathrm{d}\tilde{\phi}, \tag{23}$$

$$\int_{A_{nk}} \tilde{\phi} \cos\theta \mathrm{d}\tilde{A} = \oint_{C_{nk}} -\frac{\tilde{\phi}}{2} [\sin\theta\cos\theta + \theta] \, \mathrm{d}\tilde{\phi}, \tag{24}$$

respectively, where $C_{nk}$ is the counterclockwise path around the region $A_{nk}$.

Based on the author's exploration of the SCRIP source code and the CDO source code, the formulations correspond to Eqs. (19) and (21) being implemented.

However, the remapping weight for the longitudinal direction determined in Eq. (21) is invalid, which lacks an important characteristics of Eqs. (15) and (16). As explained, equations of the remapping weights should hold for any longitudinal origin, which means that the remapping weights $w_{3nk}^{\mathsf{ORG}}$ (Eq. 20) and $\tilde{w}_{3nk}^{\mathsf{ORG}}$ (Eq. 21) must be identical. It is demonstrated below that this characteristics is not guaranteed on these formulations.

Substituting $\tilde{\phi} = \phi - \phi_{\mathsf{ofs}}$ into Eq. (21), $\tilde{w}_{3nk}$ can be expanded as follows:

$$\tilde{w}_{3nk}^{\mathsf{ORG}} = \frac{1}{A_k} \int_{A_{nk}} (\phi - \phi_{\mathsf{ofs}}) \cos\theta \, \mathrm{d}A - \frac{w_{1nk}}{A_n} \int_{A_n} (\phi - \phi_{\mathsf{ofs}}) \cos\theta \, \mathrm{d}A$$

$$= w_{3nk}^{\mathsf{ORG}} - \frac{\phi_{\mathsf{ofs}}}{A_k} \left[ \int_{A_{nk}} \cos\theta \, \mathrm{d}A - \frac{A_{nk}}{A_n} \int_{A_n} \cos\theta \, \mathrm{d}A \right]. \tag{25}$$

Therefore, the bracket terms in Eq. (25) must be zero in order to satisfy the condition $\tilde{w}_{3nk} \equiv w_{3nk}$ for arbitrarily chosen $\phi_{\mathsf{ofs}}$. As long as the area element $\mathrm{d}A$ is a function of $\theta$ such as on the spherical coordinate system, the term $\cos\theta$ cannot be taken out of the integral, and thus the terms in the bracket are not cancelled. Equation (25) is satisfied only when $\phi_{\mathsf{ofs}} = 0$, which means that it is definitely inconsistent with its former derivation as Eqs. (15) and (16).

This inconsistency originates from invalid derivation from Eq.(15) to Eq.(20), to substitute the reference longitude following Eq. (18). While transformation of the position vector into the spherical coordinates is conducted on Eq. (7) in J99, the same




procedure should be conducted on Eq. (6) instead, because the position vector cannot be extracted from the integral on the spherical coordinate.

A new symbol is introduced to designate such a reference point as $\mathbf{p}_n = [\theta_p, \phi_p]^\mathsf{T}$, hereafter symbolically referred to as *pivot*, in order to distinguish it from the centroid described above. Then the condition of the reference coordinate $(\theta_p, \phi_p)$, including the metric scale factor, are formulated instead of Eqs. (17) and (18) as follows:

$$\int_{A_n} \theta_p \, \mathrm{d}A = \int_{A_n} \theta \, \mathrm{d}A, \tag{26}$$

$$\int_{A_n} \phi_p \cos\theta \, \mathrm{d}A = \int_{A_n} \phi \cos\theta \, \mathrm{d}A. \tag{27}$$

The elements $\theta_p$ and $\phi_p$ can be extracted from the integral, and the reference coordinates are formulated as:

$$\theta_p = \frac{1}{A_n} \int_{A_n} \theta \, \mathrm{d}A, \tag{28}$$

$$\phi_p = \left( \int_{A_n} \phi \cos\theta \, \mathrm{d}A \right) \bigg/ \left( \int_{A_n} \cos\theta \, \mathrm{d}A \right). \tag{29}$$

While the formulation in the latitudinal direction is identical (Eq. 17 and 28), that in the longitudinal direction is different regarding the treatment of $\cos\theta$ term in the denominator (Eq. 18 and 29).

The remapping weight for the longitudinal direction are reformulated as:

$$w_{3nk\mathsf{P}} = \frac{1}{A_k} \int_{A_{nk}} (\phi - \phi_p) \cos\theta \, \mathrm{d}A = \frac{1}{A_k} \int_{A_{nk}} \phi \cos\theta \, \mathrm{d}A - \frac{1}{A_k} \frac{\Omega_{3nk}}{\Omega_{3n}} \int_{A_n} \phi \cos\theta \, \mathrm{d}A, \tag{30}$$

$$\Omega_{3nk} = \int_{A_{nk}} \cos\theta \mathrm{d}A. \tag{31}$$

Equation (30) is confirmed to hold even with substituting $\tilde{\phi} = \phi - \phi_{\mathsf{ofs}}$, as follows:

$$\tilde{w}_{3nk\mathsf{P}} = \frac{1}{A_k} \int_{A_{nk}} (\phi - \phi_{\mathsf{ofs}}) \cos\theta \, \mathrm{d}A - \frac{1}{A_k} \frac{\Omega_{3nk}}{\Omega_{3n}} \int_{A_n} (\phi - \phi_{\mathsf{ofs}}) \cos\theta \, \mathrm{d}A, \tag{32}$$

$$= w_{3nk\mathsf{P}} - \frac{\phi_{\mathsf{ofs}}}{A_k} \left[ \int_{A_{nk}} \cos\theta \, \mathrm{d}A - \frac{\Omega_{3nk}}{\Omega_{3n}} \int_{A_n} \cos\theta \, \mathrm{d}A \right],$$

where the bracket terms are cancelled with definition of Eq. (31).

How the remapping weights are influenced by the invalid formulation can be demonstrated by using a simple configuration in which both source and destination grids are set as RLL grids on a unit sphere, and the cells are equally spaced along the longitude and latitude. The latitudes and longitudes of the grid lines (cell corner coordinates) are expressed as $\theta = 180° (j - N_\theta/2)/N_\theta, j = 0, \cdots, N_\theta$; and $\phi = 360° (i - 0.5 - N_\phi/2)/N_\phi, i = 0, \cdots, N_\phi$, respectively. The source and destination grids adopt $N_\theta, N_\phi = 64, 128$ and $N_\theta, N_\phi = 128, 256$, respectively, where a source cell contains $2 \times 2 = 4$ destination



cells, and a destination cell does not extend over multiple source cells. Figure 1 shows the distribution of the remapping weight $w_{3nk}$ over the example source/destination configuration. One source cell has four remapping weights for each overlapped
destination cell; those for the north-west designation cells are plotted in the figure. (It is for this reason that the figure is not symmetric about the equator.) Since the relative orientation of a source cell and its overlapped destination cells is equivalent along the longitudinal direction, the remapping weight must be axisymmetric. Figure 1(a) displays the results for weights computed with the original formulation Eq. (20), clearly showing the breaking of symmetry. In contrast, in Fig 1(b), the remapping weights were computed using the formulation satisfying the pivot condition (Eq. 30) which produces the axisymmetric results
shown in the figure.

In Fig. 1, computation of remapping weights is conducted not with the formulations using longitude adjustment for each source cell (Eq. 21 and 32), but with those using the globally-fixed longitude origin. As described above, J99 suggests to adopt the source grid cell center as reference longitude for each source cell, therefore the breaking of symmetry away from $\phi = 0$ shown in Fig. 1(a) is practically not applicable. In fact, this suggestion of adjustment in longitude in the original algorithm
minimizes or even erases all the problem as a side effect.

Formally, it is possible to substitute $\phi_{\mathsf{ofs}} = \phi_p$ in Eq. (25):

$$\underset{\mathsf{ORG}}{\tilde{w}_{3nk}} = \frac{1}{A_k} \int_{A_{nk}} (\phi - \phi_p) \cos\theta \, \mathrm{d}A - \frac{w_{1nk}}{A_n} \int_{A_n} (\phi - \phi_p) \cos\theta \, \mathrm{d}A. \tag{33}$$

Introducing the condition of $\phi_p$ in Eq. (29), the integral part in the second term of Eq. (33) unconditionally becomes zero. Although the second term is inconsistent overall, it is shown that only the coefficient makes the term inconsistent by comparison
between Eq. (21) and (30). Thus the cancellation of the second term in Eq. (33) provides identical solutions with Eq. (30), even though the coefficients are invalid, since these coefficients are essentially erased by the zero-valued integration term. The problem is, of course, that it does not make sense to expect a valid $\phi_p$ based on an inconsistent computation of the remapping weights.

However, there is, indeed, an explanation.

Although not forced, it is quite natural to set the offset longitude $\phi_{\mathsf{ofs}}$ as the center of the longitude range of the source grid cells. One reason for this is that a sample program included in SCRIP makes the computation this way; another is that the center longitude is often used for other situations, e.g., visualization, and thus they can be easily prepared. For some special cases, such as benchmark tests, the central longitude is used to evaluate the flux gradient, which is not generally possible for practical applications. For the RLL rectangle grid cells in spherical coordinates, the center longitude is identical to the pivot longitude,
and therefore the rotation helps to cancel the contribution of the pivot term. Moreover, if a cell is symmetric along a meridian, then, naturally, the pivot coordinate coincides with the center longitude. In most cases using various shapes of grid cells, the center longitude defined by the user for particular target grid cells may not be far from the pivot longitude, and the problem of the incorrect contribution of the pivot term can be rendered insignificant, as shown in the previous section. In principle, the offset longitude is left to the user's discretion, and these side effects are generally unexpected.

According to the CDO reference manual, the second-order conservative remapping command (`REMAPCON2`) is not available for unstructured source grids. This is good news for CDO users. Thanks to the mid-longitude offsetting, there may be no risk



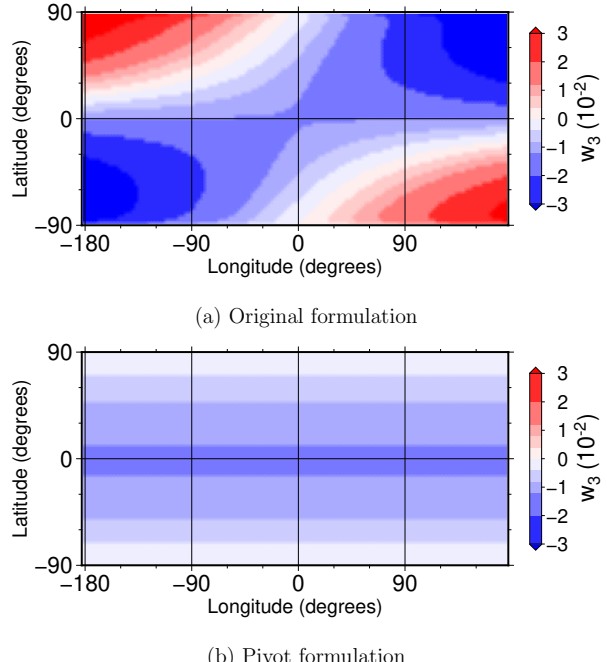

(a) Original formulation

(b) Pivot formulation

**Figure 1.** Demonstration of the remapping weight computation. (a) $w_{3\,nk}$ using the J99 original algorithm (Eq. 20) (b) $w_{3\,nk}$ by Eq. (30).

that the user will suffer from the inconsistent formulation of the remapping weights. However, the author is not fully convinced, and such a conclusion should be confirmed by an expert in the area.

The correction presented above is one possible solution for consistency. For example, a different procedure to formulate the remapping coefficients may be branched from Eq. (10). Given a constant derivative across source grid cell instead of gradient, Eq. (10) may be reformulated as follows:

$$f_n = \overline{f}_n + \left(\frac{\partial f}{\partial \theta}\right)_n (\theta - \theta_c) + \left(\frac{\partial f}{\partial \phi}\right)_n (\phi - \phi_c), \tag{34}$$

where the $\cos\theta$ term in the longitudinal direction is cancelled. Using this flux approximation, the remapping weight in the longitudinal direction (Eq. 15) is replaced as:

$$w_{3\,nk\mathsf{C}} = \frac{1}{A_k} \int_{A_{nk}} (\phi - \phi_c)\,\mathrm{d}A. \tag{35}$$

Since the term $\cos\theta$ is not included, the constraint of reference longitude in Eq. (6) can be replaced with a simple formulation as:

$$\int_{A_n} \phi\,\mathrm{d}A - \int_{A_n} \phi_c\,\mathrm{d}A = 0. \tag{36}$$



Finally, the remapping weight is formulated as:

$$\tilde{w}_{3nk\mathsf{C}} = \frac{1}{A_k} \int\limits_{A_{nk}} \tilde{\phi}\, \mathrm{d}\tilde{A} - \frac{w_{1nk}}{A_n} \int\limits_{A_n} \tilde{\phi}\, \mathrm{d}\tilde{A}. \tag{37}$$

Again, Eq. (37) is valid for any longitude origin (not shown because it is trivial). Formally, the formulation of Eq. (37) is similar to one on the Cartesian coordinate except for the area element.

## 3 Experiment and discussion

In order to demonstrate the argument of the present paper described in the previous section, a series of sensitivity experiments are performed. The main focus of the present paper is to show how J99 and SCRIP are influenced by inconsistent reference longitudes. The evaluation of them among the other remapping packages is far beyond the scope.

All the remapping experiments are performed using SCRIP version 1.5, with minimum necessary modification relating to the remapping weight computation. The version with proposed modification is hereafter referred to as SCRIP-p to distinguish it from the official SCRIP.

The offset longitude is specified in the external input file in original SCRIP application, which is also followed by SCRIP-p. The longitude adjustment with the externally prescribed offset longitude is left as is, since it is, in any event, necessary in order to deal with the periodic boundary condition in a simple way.

It is worth mentioning that the sensitivity to the offset longitude can be examined only by replacing the values in the input data (variable `src_grid_centroid_lon` in the input file) Either the source code of the program to compute remapping weights or that to perform remapping can be used without any modification, even for the original SCRIP implementation. Although test program included in the official SCRIP uses the offset longitude for computing the input source field to remapping, the input field is also prescribed by external files in the present study thus the offset longitude is not used anywhere except for the remapping weight computation.

There may be other problems in J99 and original SCRIP: it is reported that the treatment of parametric form for cell sides in the algorithm results in inaccuracies at intersection computation (J99, Jones, 2024). As noted, the main focus of the present study is not to improve the algorithm, but rather to report how the inconsistencies influence the performance in the past application. Therefore all the program source codes, except for those related to the inconsistent reference longitude, are left as they were. The original variable names are left as they were even if they contain the word `centroid` as a misconception, in order to minimize the modification.

## 3.1 Summary of original and corrected formulations

Two remedies are introduced in the previous section, in order to preserve consistency during the derivation. The original formulation as well as the two alternate formulations are summarised here for reference. The first-order remapping weight, $w_{1nk}$, and the second-order remapping weight in the latitudinal direction, $w_{2nk}$, are identical to those originally presented in J99 but are listed here for completeness.





The methods presented below are called 'Schemes', however, they are no more than correction to the original method. They are not new algorithm for the second-order conservative remapping, but rather minor variation of the original algorithm to share most of the equations for remapping weights except for the final formulation.

### 3.1.1  Scheme N — original (*native*) method

Scheme N is the implementation of the original J99 formulation, which adopts $(\theta_n, \phi_n)$ formulation (Eqs. 17, 18) as the
reference point, with introducing the relative longitude to an offset longitude $\phi_{\mathsf{ofs}}$. The flux approximation is formulated as

$$\overline{F}_k = \sum_{n=1}^{N} \left[ \overline{f}_n w_{1\,nk\mathsf{N}} + \left( \frac{\partial f}{\partial \theta} \right)_n w_{2\,nk\mathsf{N}} + \left( \frac{1}{\cos\theta} \frac{\partial f}{\partial \phi} \right)_n w_{3\,nk\mathsf{N}} \right], \tag{38}$$

and the corresponding remapping weights are formulated as

$$\begin{cases} w_{1\,nk\mathsf{N}} = \dfrac{1}{A_k} \displaystyle\int\limits_{A_{nk}} \mathrm{d}A, \\[2mm] w_{2\,nk\mathsf{N}} = \dfrac{1}{A_k} \displaystyle\int\limits_{A_{nk}} (\theta - \theta_n)\,\mathrm{d}A = \dfrac{1}{A_k} \displaystyle\int\limits_{A_{nk}} \theta\,\mathrm{d}A - \dfrac{w_{1\,nk}}{A_n} \displaystyle\int\limits_{A_n} \theta\,\mathrm{d}A, \\[2mm] w_{3\,nk\mathsf{N}} = \dfrac{1}{A_k} \displaystyle\int\limits_{A_{nk}} (\tilde\phi - \tilde\phi_n)\cos\theta\,\mathrm{d}\tilde{A} = \dfrac{1}{A_k} \displaystyle\int\limits_{A_{nk}} \tilde\phi\cos\theta\,\mathrm{d}\tilde{A} - \dfrac{w_{1\,nk}}{A_k} \displaystyle\int\limits_{A_n} \tilde\phi\cos\theta\,\mathrm{d}\tilde{A}, \\[2mm] \tilde\phi = \phi - \phi_{\mathsf{ofs}}. \end{cases} \tag{39}$$

As described in the previous section, the formulation of Scheme N is valid only when the offset longitude $\phi_{\mathsf{ofs}}$ equals to the
pivot longitude $\phi_p$ of the source cell $n$, defined in Eq. (29). In this particular case, the pivot term contributes virtually nothing to the remapping weights.

### 3.1.2  Scheme P — *pivot* method

Scheme P is a mostly straightforward implementation of the original J99 formulation, where only the invalid computation of the remapping weight $w_{3\,nk}$ is replaced according to the pivot condition. It adopts $(\theta_p, \phi_p)$ formulation (Eqs. 28, 29) as
the reference point. Formally, the centroid definition must be excluded from the beginning of the implementation as it is incompatible with this formulation. The flux approximation is formulated as

$$\overline{F}_k = \sum_{n=1}^{N} \left[ \overline{f}_n w_{1\,nk\mathsf{P}} + \left( \frac{\partial f}{\partial \theta} \right)_n w_{2\,nk\mathsf{P}} + \left( \frac{1}{\cos\theta} \frac{\partial f}{\partial \phi} \right)_n w_{3\,nk\mathsf{P}} \right], \tag{40}$$



and the corresponding remapping weights are formulated as

$$
\begin{cases}
w_{1\,nk\text{P}} = \dfrac{1}{A_k} \displaystyle\int_{A_{nk}} \mathrm{d}A, \\[2mm]
w_{2\,nk\text{P}} = \dfrac{1}{A_k} \displaystyle\int_{A_{nk}} (\theta - \theta_p)\,\mathrm{d}A = \dfrac{1}{A_k} \displaystyle\int_{A_{nk}} \theta\,\mathrm{d}A - \dfrac{w_{1\,nk}}{A_n} \displaystyle\int_{A_n} \theta\,\mathrm{d}A, \\[2mm]
w_{3\,nk\text{P}} = \dfrac{1}{A_k} \displaystyle\int_{A_{nk}} (\phi - \phi_p)\cos\theta\,\mathrm{d}A = \dfrac{1}{A_k} \displaystyle\int_{A_{nk}} \phi\cos\theta\,\mathrm{d}A - \dfrac{1}{A_k}\dfrac{\Omega_{3\,nk}}{\Omega_{3\,n}} \displaystyle\int_{A_n} \phi\cos\theta\,\mathrm{d}A, \\[2mm]
\Omega_{3\,nk} = \displaystyle\int_{A_{nk}} \cos\theta\,\mathrm{d}A.
\end{cases}
\tag{41}
$$

The new term $\Omega_{3\,nk}$ to be applied in the evaluation of $w_{3\,nk}$ is introduced. This term is not a remapping weight but is computed with the same procedure as the other three remapping weights. The integral part of $\Omega_{3\,nk}$ is computed by transforming it into a line integral using Gauss's divergent theorem following the J99 method for the other integrals, and is formulated as

$$
\int_{A_{nk}} \cos\theta\,\mathrm{d}A = \oint_{C_{nk}} -\frac{\sin\theta\cos\theta + \theta}{2}\,\mathrm{d}\phi.
\tag{42}
$$

Although the replacement involves only the computation of a single variable, the source code modification would be the most substantial among the three proposals since the treatment of additional variable $\Omega_{3\,nk}$ must be introduced concurrently with the three standard weights.

The formulation of scheme P is consistent for any longitude origin, thus it is not necessary to introduce the offset longitude, while it is still valid with any offset longitudes. Also, Schemes O and P produces the identical solution when the offset longitude $\phi_{\text{ofs}}$ equals to $\phi_p$ in Eq. (39).

### 3.1.3 Scheme C — *centroid* method

Scheme C is the implementation that deviates from the original J99 formulation, whereby the modification is introduced after the definition of the centroid (Eq. 34). Formally, maintaining the centroid formulation, the flux approximation must be reformulated. It adopts $(\theta_c, \phi_c)$ formulation (Eqs. 17, 36) as the reference point.

The flux approximation is replaced with the formulation

$$
\overline{F}_k = \sum_{n=1}^{N} \left[ \overline{f}_n w_{1\,nk\text{C}} + \left(\frac{\partial f}{\partial \theta}\right)_n w_{2\,nk\text{C}} + \left(\frac{\partial f}{\partial \phi}\right)_n w_{3\,nk\text{C}} \right],
\tag{43}
$$



and the corresponding remapping weights are formulated as

$$
\begin{cases}
w_{1\,nk\mathtt{C}} = \dfrac{1}{A_k} \displaystyle\int\limits_{A_{nk}} \mathrm{d}A, \\[2ex]
w_{2\,nk\mathtt{C}} = \dfrac{1}{A_k} \displaystyle\int\limits_{A_{nk}} (\theta - \theta_c)\,\mathrm{d}A = \dfrac{1}{A_k} \displaystyle\int\limits_{A_{nk}} \theta\,\mathrm{d}A - \dfrac{w_{1\,nk}}{A_n} \displaystyle\int\limits_{A_n} \theta\,\mathrm{d}A, \\[2ex]
w_{3\,nk\mathtt{C}} = \dfrac{1}{A_k} \displaystyle\int\limits_{A_{nk}} (\phi - \phi_c)\,\mathrm{d}A = \dfrac{1}{A_k} \displaystyle\int\limits_{A_{nk}} \phi\,\mathrm{d}A - \dfrac{w_{1\,nk}}{A_n} \displaystyle\int\limits_{A_n} \phi\,\mathrm{d}A.
\end{cases}
\tag{44}
$$

This method involves not only a modification of the remapping weight computation, but requires that the caller replace the gradient (including the cosine latitude factor) of the derivative. It is posited that this method is the easiest to extend to a higher-order conservative remapping scheme since the formulation should be mostly compatible with the Cartesian formulation.

The integral part of $w_{3\,nk}$ is computed using Gauss's divergent theorem as follows:

$$
\int\limits_{A_{nk}} \phi\,\mathrm{d}A = \oint\limits_{C_{nk}} -\phi\sin\theta\,\mathrm{d}\phi.
\tag{45}
$$

Similar to scheme P, the formulation of scheme C is consistent for any longitude origin,

### 3.2 Configuration of experiments

In the present study, only the domain of the RLL grid on a unit sphere of $N_\theta$ latitudes and $N_\phi$ longitudes, both for the source and destination grids, is examined. The latitudes and longitudes of the grid lines (cell corner coordinates) are expressed as $\theta = 180^\circ\,(j - N_\theta/2)$, $j = 0, \cdots, N_\theta$; and $\phi = (i/N_\phi)$, $i = 0, \cdots, N_\phi$, respectively (this is slightly different from the domain definition used for the demonstration in Fig. 1, which does not influence the discussion). The size of the source grid cell is set as $(N_\theta, N_\phi) = (1024, 2048)$. Several destination grid sizes are examined, including $(N_\theta, N_\phi) = (90, 180)$, $(180, 360)$, $(360, 720)$, $(720, 1440)$. Three idealized experiments A1, A2 and A3 were conducted following J99 and Mahadevan et al. (2022). A1 and A2 correspond to ANALYTICALFUN1 and ANALYTICALFUN2 presented in Mahadevan et al. (2022), respectively. A2 also corresponds to that of the experiments in presented in J99 whose source field is named as $Y_2^2$. A3 corresponds to another experiment presented in J99, named as $Y_{32}^{16}$. A2 and A3 also appear in Lauritzen and Nair (2008); Ullrich et al. (2009).

The source field in experiment A1 is a combination of spherical harmonics functions with frequency wave similar to order 3, given by

$$
\psi = Y_3^2 + Y_3^3,
\tag{46}
$$

where $Y_m^l$ represents the real spherical harmonic functions evaluated for degree $m$ and polynomial order $l$.

In experiments A2 and A3, a relatively smooth function resembling a spherical harmonic of order 2 and azimuthal wavenumber 2 (named as $Y_2^2$),

$$
\psi = 2 + \cos^2\theta\,\cos(2\phi),
\tag{47}
$$



and a relatively high-frequency wave similar to a spherical harmonic of order 32 and azimuthal wavenumber 16 (named as $Y_{32}^{16}$),

$$\psi = 2 + \sin^{16}(2\theta)\cos(16\phi), \tag{48}$$

are used as input for the source grid in each experiment. The mid-longitude and mid-latitude coordinates for each cell are used
as a reference point to compute $\psi$ and its gradient to input.

    All the experiments are conducted using the test program included in the official SCRIP package with minimum modification. It is worth mentioning that a special treatment for elements around the poles is implemented in the official package, which is switched off in the present study.

    The performance of the conservative remapping algorithms was evaluated using Metrics for Intercomparison of Remapping
Algorithms (MIRA) package (Guerra et al., 2021; Mahadevan et al., 2022), with help of TempestRemap (Ullrich and Taylor, 2015; Ullrich et al., 2016) that is conducted to prepare the input fields. Several measures are available by MIRA. Global conservation properties are evaluated using $L_g$, which corresponds to relative change in the global integral of the scalar field value on the source and the destination grids. The standard accuracy measures, $\|E\|_{L_2}$ and $\|E\|_{L_\infty}$ are presented, which correspond to those used the second-order norm $l_2$ and the infinity norm $l_\infty$, respectively. A gradient preservation measure
$\|E\|_{H_1}$ computed by MIRA package is also presented. The explicit definition of these norms are presented in Mahadevan et al. (2022).

    As shown in Sect. 2.2, it is speculated that for RLL rectangular cell cases, the offset longitude virtually works as the pivot longitude ($\phi_p$ in Eq. 29) in the official SCRIP, which would erase the fundamental problem. For general shapes of grid cells, the offset longitude may not be the same as the pivot longitude. To investigate the sensitivity of this deviation, a simple experiment
is presented using the official SCRIP implementation.

    In the official implementation, the mid-longitude for each cell is introduced for the offset:

$$\phi_{\mathsf{ofs}} = \frac{\phi_0 + \phi_1}{2}, \tag{49}$$

where $\phi_0$, $\phi_1$ are the longitude boundaries of the source cell. Using this offset to keep the difference in longitudes within $360°$, it is easy to avoid the multiple-value longitude issues. Conversely, the offset can be anywhere as far as it is sufficient to avoid
the multiple-value issues,

    In order to demonstrate the present paper's argument, three sensitivity experiments are performed: the first one is control case, to adopt mid-longitude (Eq. 49) for each cell. The second one is *cell-edge* case, in which the offset matches the boundary for each cell (i.e., $\phi_{\mathsf{ofs}} = \phi_1$). The third one is *global* case, in which the offset longitude is set as $\phi_{\mathsf{ofs}} \equiv 180°$ for all the source grid cells[1].

Since the pivot longitude matches the mid-longitude of the RLL rectangle cell, the same results should be obtain by Scheme O and P in the control case. The second experiment practically corresponds to an extreme case. It can be naturally expected

---

[1]There is a limit in the SCRIP-p implementation in this impractical case: the longitude relative to the offset must be within $-180°$ to $180°$; to work correctly, this is the only choice. For normal applications, this causes no problem.



that the pivot longitude is within the cell for general shapes of the source grid cell, therefore this can be a maximum difference of the pivot and offset longitudes for usual application. The third experiment is more than an extreme case where the equations really hold true while it may be rare for typical SCRIP application.

## 3.3 Results

Since the algorithm discussed in the present paper is conservative remapping, it is important to check the errors in the global conservation for all the experiment. Tables 1 is the summary the metric $L_g$, obtained by sensitivity experiments A2. The results of other two experiments A1 and A3 are in the supplement. All norms are computed using the result of one-time forward remapping from the source grid to four variation of destination grid. The metrics in the first row in the tables (Scheme O and offset mid) correspond to those obtained by the official SCRIP. These are reference values of the present study, and evaluation of metrics are examined relative to these values. With floating-point arithmetic of binary64 (specified in IEEE 754-2008 standard, usually referred to as *double-precision*), we have around 15 significant digits.

Except for the Scheme N, global-offset cases, the global conservation properties after one-time remapping mostly agreed to one part in the first 15 digits, thus the remapping are conservative to machine accuracy, which is the same conclusion as presented in J99. For the Scheme N, global-offset cases, errors in the global conservation are prominent among the experiments. As far as the multiple-value problem is avoided, the remapping results should be insensitive to the choice of offset longitude. Thus it is confirmed that the formulation of the reference (centroid) term in the original algorithm is invalid and have a potential to damage the important properties. However, the global-offset configuration is practically more than extreme which may never happen in the typical application. Instead, Scheme N, cell-edge cases are regarded as an extreme case. Global conservation obtained by Scheme N, cell edge cases are comparable to mid-edge cases, thus no significant damages on the conservation are expected with the original algorithm.

Mathematically, Schemes P and C should give identical results by replacing the offset longitude, but it is not presented in the experiments. This is due to the finite-precision arithmetic. For example, difference in longitude is not computed in degree units, but in radian units after degree-to-radian conversion in the original implementation of SCRIP, which may result in slightly non-uniform values. However, both Schemes P and C show comparable errors in the global conservation even for global-edge cases, which confirms the expected insensitivity on the offset longitude.

Tables 2, 3, 4 are the summary the metric $\|E\|_{L_2}$, $\|E\|_{L_\infty}$, $\|E\|_{H_1}$, respectively, obtained by sensitivity experiments A2, one-time forward remapping. The results of other two experiments A1 and A3 are summarised in the Supplement. In the tables, the first seven digits are shown for comparison. In general, all three experiments show qualitatively similar results.

As shown in the tables, the metrics of cell-edge case are slightly deviated from those corresponding mid-cell case. Difference relative to the mid-cell case increases according to the increase in resolution of the destination grid, however, the metric mostly maintains its order of magnitude. Since the cell-edge case is regarded as an extreme for practical application, the field after remapping may remain similar without a significant impact as far as the offset longitude is within a source cell.

It is clearly shown that the metrics of global-offset case changes their order of magnitudes. Difference relative to the mid-cell case increases according to the increase in resolution of the destination grid, and they reach around 1000-times larger value





**Table 1.** Summary of results of the sensitivity experiments A2 for the offset longitude using Schemes N, P, C. The second column indicates the offset longitude, where mid, edge, global correspond to the $\phi_{\text{ofs}} = (\phi_0 + \phi_1)/2$, $\phi_{\text{ofs}} = \phi_1$, and $\phi_{\text{ofs}} = 180°$ cases, respectively. Destination grid sizes are $(N_\theta, N_\phi) = (90, 180), (180, 360), (360, 720), (720, 1440)$. The absolute value of relative error in global conservation ($|L_g|$, see Mahadevan et al., 2022 for its definition) is shown in table.

| Scheme | Offset | $(90, 180)$ $[\times 10^{-15}]$ | $(180, 360)$ $[\times 10^{-15}]$ | $(360, 720)$ $[\times 10^{-15}]$ | $(720, 1440)$ $[\times 10^{-15}]$ |
|---|---|---|---|---|---|
| N | mid | 0.989 | 2.261 | 0.565 | 1.272 |
| N | edge | 0.706 | 2.120 | 0.141 | 1.272 |
| N | global | 3199. | 7960. | 1990. | 498.4 |
| P | mid | 0.989 | 1.837 | 0.706 | 1.554 |
| P | edge | 0.989 | 2.120 | 0.848 | 1.554 |
| P | global | 0.706 | 1.979 | 0.989 | 1.554 |
| C | mid | 0.848 | 1.979 | 0.565 | 1.554 |
| C | edge | 0.706 | 2.261 | 0.565 | 1.554 |
| C | global | 0.565 | 1.979 | 0.565 | 1.554 |

for the highest resolution in the present study. Although the magnitudes of these metrics may be still reasonably small, the significant sensitivity on the metrics to the offset longitudes confirms the statement of the present study, which the formulation of the reference (centroid) term is invalid.

The metrics obtained by Scheme P, mid-cell case show identical results with those obtained by Scheme O as expected, because the pivot longitudes match the mid-cell longitudes. Due to influence from cancellation and rounding-off during the floating point computation, the cell-edge and global-offset cases show different metrics from the mid-cell case. Nevertheless, all of them are preserved significantly better than those of Scheme O. Even in the global-offset cases, magnitude of all the metrics are unaffected. Therefore, the formulation of Scheme P is expected as the valid correction of the second-order conservative remapping scheme of J99.

The metrics obtained by Scheme C, another candidate of the correction, show similar results to those by Scheme P. For lower resolution of the destination grid, the metrics are mostly the same as Scheme P, while for higher resolution, the metrics are slightly larger than those of Scheme P. Among three metrics, a gradient preservation measure $\|E\|_{H_1}$ is mostly affected, which probably reflects the difference in the formulation of the flux approximation in terms of gradients.

       In order to quantify the stability of remapping, iterative two-way remapping (i.e., sequence of forward and backward remap-
ping) is conducted. In the case of Scheme N with global-offset, the iterate remapping is extremely unstable, where the metrics explode within first twenty steps. On the other hand, the other two Schemes P and C show stable behaviour where the difference in metric among three variation of offsets are hardly visible on the plots. This result also supports the argument of the present study that the algorithm should hold for any longitudinal origin. The results of three experiments after 1000-time iterate remapping are summarised in the Supplement.



**Table 2.** The same as Tab. 1 but $\|E\|_{L_2}$ of Experiment A2 is shown.

| Scheme | Offset | $(90, 180)$ $[\times 10^{-5}]$ | $(180, 360)$ $[\times 10^{-6}]$ | $(360, 720)$ $[\times 10^{-7}]$ | $(720, 1440)$ $[\times 10^{-7}]$ |
|---|---|---|---|---|---|
| N | mid | 1.098306 | 2.683551 | 6.071390 | 1.194096 |
| N | edge | 1.098307 | 2.683582 | 6.073436 | 1.329164 |
| N | global | 1.356547 | 15.13717 | 182.9107 | 677.0403 |
| P | mid | 1.098306 | 2.683551 | 6.071390 | 1.194096 |
| P | edge | (11d) | (11d) | (10d) | (11d) |
| P | global | (11d) | (11d) | (10d) | (11d) |
| C | mid | 1.098306 | 2.683551 | 6.071393 | 1.198306 |
| C | edge | (12d) | (11d) | (9d) | (8d) |
| C | global | (11d) | (11d) | (9d) | (10d) |

**Table 3.** The same as Tab. 1 but $\|E\|_{L_\infty}$ of Experiment A2 is shown.

| Scheme | Offset | $(90, 180)$ $[\times 10^{-5}]$ | $(180, 360)$ $[\times 10^{-6}]$ | $(360, 720)$ $[\times 10^{-7}]$ | $(720, 1440)$ $[\times 10^{-7}]$ |
|---|---|---|---|---|---|
| N | mid | 1.677407 | 4.105416 | 9.388161 | 2.401264 |
| N | edge | 1.677408 | 4.106974 | 9.410346 | 2.510583 |
| N | global | 2.694874 | 46.92178 | 568.6044 | 2115.962 |
| P | mid | 1.677407 | 4.105416 | 9.388161 | 2.401264 |
| P | edge | (10d) | (8d) | (7d) | (15d) |
| P | global | (9d) | (8d) | (8d) | (8d) |
| C | mid | 1.677407 | 4.105464 | 9.388614 | 2.401393 |
| C | edge | (15d) | (8d) | (8d) | (15d) |
| C | global | (9d) | (15d) | (8d) | (9d) |

435 Figure 2 show the evolution of the metric $\|E\|_{L_2}$, $\|E\|_{L_\infty}$, $\|E\|_{H_1}$, respectively, along iterative remapping obtained by sensitivity experiments A2. The results of other two experiments A1 and A3 are summarised in the Supplement. In general, all three experiments show qualitatively similar results also for the iterative remapping. The evolution of metrics of cell-edge case mostly overlaps those of mid-cell case for Scheme N, except for the high resolution destination grid. In experiment A2, the results of destination grid size as $(720, 1440)$ clearly deviates along the iteration steps, due to accumulation of small difference

440 in the metrics. Practically, even for general shapes of source grid cells, the mid-cell longitude would be well close to the pivot longitude, thus no critical impact on the remapping may be expected. The difference in metrics evolution between Scheme N, mid-cell case and Scheme P is minor, whose relative difference is below around $10^{-7}$. It can be concluded that for simple application as RLL where the mid-cell longitude matches the pivot longitude for each source cell, a sufficiently reasonable





**Table 4.** The same as Tab. 1 but $\|E\|_{H_1}$ of Experiment A2 is shown.

| Scheme | Offset | $(90, 180)$ $[\times 10^{-5}]$ | $(180, 360)$ $[\times 10^{-5}]$ | $(360, 720)$ $[\times 10^{-6}]$ | $(720, 1440)$ $[\times 10^{-6}]$ |
|---|---|---|---|---|---|
| N | mid | 4.776152 | 1.173408 | 3.381292 | 9.707323 |
| N | edge | 4.776**172** | 1.17**5321** | 3.**696067** | **11.67384** |
| N | global | **16.95678** | **77.74871** | **1731.057** | **7520.313** |
| P | mid | 4.776152 | 1.173408 | 3.381292 | 9.707323 |
| P | edge | *(11d)* | *(10d)* | *(8d)* | *(8d)* |
| P | global | *(10d)* | *(10d)* | *(8d)* | *(7d)* |
| C | mid | 4.776152 | 1.173418 | 3.383763 | 9.928686 |
| C | edge | *(10d)* | *(11d)* | *(8d)* | *(7d)* |
| C | global | *(12d)* | *(10d)* | *(9d)* | *(8d)* |

remapping can be obtained. The difference in metrics evolution between Scheme P and C is still minor, while it sometimes

reaches to the second digits. Mostly, the metrics by P are slightly smaller than ones by C, therefore it is recommended to adopt Scheme P when a strict precision is required as much as possible.

Finally, the convergence rates for three Schemes of all the experiments are shown. These convergence rates are calculated using the script provided by MIRA Dataset (Mahadevan et al., 2021), with some minor adjustment for the present study. It is a slope obtained by the linear regression of metrics logarithms as a function of the characteristic spatial length of the destination

mesh. Following Mahadevan et al. (2022), the spatial length is defined as the inverse of the square root of number of destination grids in the present study. In Mahadevan et al. (2022) the convergence rates are computed using uniform refinements in both source and destination grids, while in the present study, only the destination grids are changed with keeping the source grid. Also, in the present study both source and destination grids are RLL, which are not discussed in Mahadevan et al. (2022). Thus the convergence rates in the present study may not be directly comparable to those in Mahadevan et al. (2022). Since the

focus of the present study is to evaluate the influence of inconsistent reference longitudes on J99 and SCRIP remapping, it is considered to be sufficient just by presenting relative performance of the corrected schemes.

Table 5 summarize the convergence rates for three Schemes of all the experiments. As noted, only relative comparison is presented here. As presented in the tables and figures above, Scheme N mid-cell and Scheme P both cases show the same convergence rates for all the experiments (Scheme P global offset cases are confirmed to be also the same, which are not

shown in the table). Scheme N cell-edge cases mostly show smaller convergence rates than corresponding mid-cell cases, except for that of the metric $\|E\|_{L_\infty}$ for experiment A1, which show a slightly larger value. Since the results of remapping should be independent, in principle, on even a small deviation of offset longitudes, it is not important whether the convergence rate is larger or not than the mid-cell case. Although the influence on the convergence rates may be reasonably small even in the practically extreme condition of offset longitudes, a visible sensitivity on the metrics again confirms the statement of the

present study, which the formulation of the reference (centroid) term is invalid.





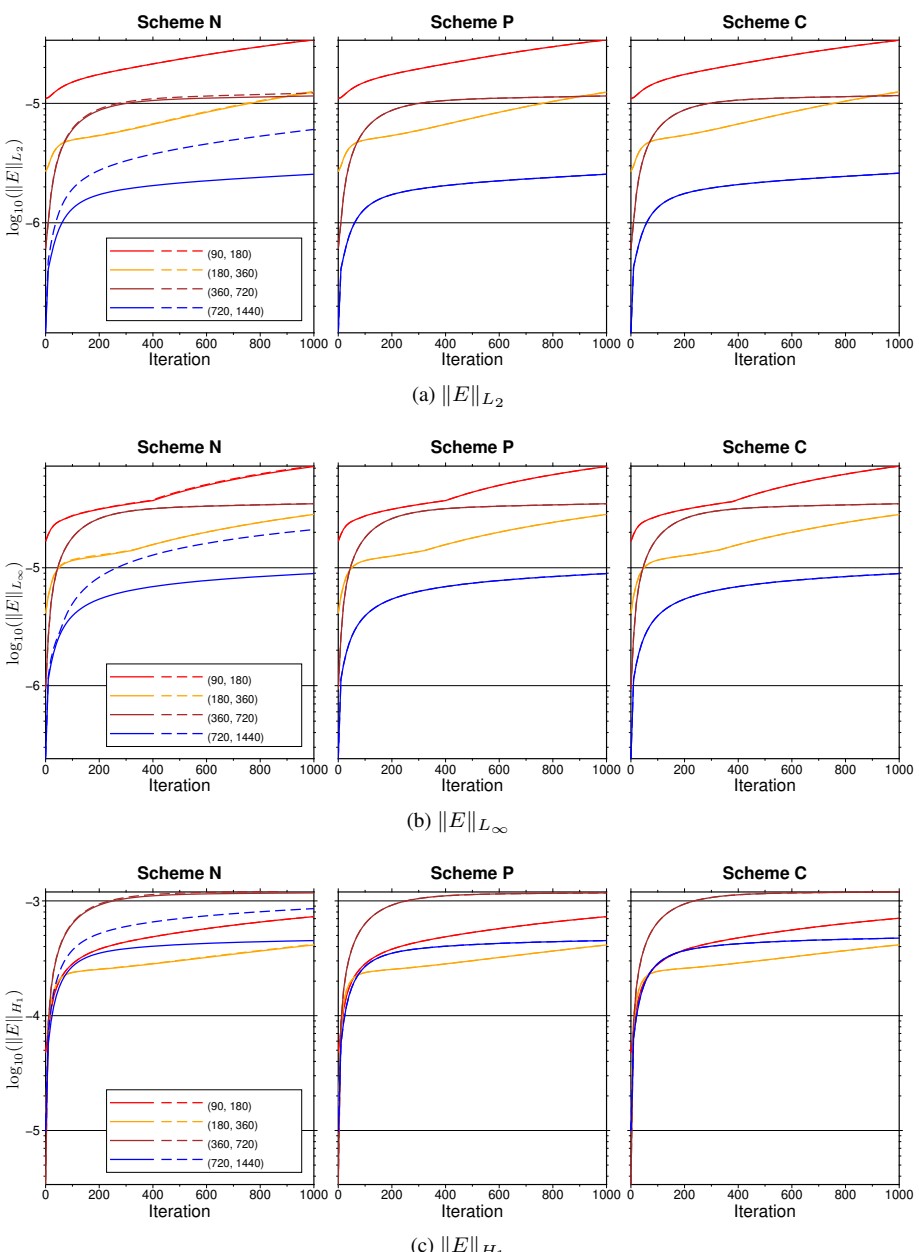

**Figure 2.** Results of the sensitivity experiments A2 for the offset longitude using Schemes N, P, C. The metrics (a) $\|E\|_{L_2}$ (b) $\|E\|_{L_\infty}$ (c) $\|E\|_{H_1}$ as functions of iteration number of two-way remapping are shown. Destination grid sizes are $(N_\theta, N_\phi) = (90, 180)$, $(180, 360)$, $(360, 720)$, $(720, 1440)$. Solid and dashed lines indicate that $\phi_{\mathsf{ofs}} = (\phi_0 + \phi_1)/2$ and $\phi_{\mathsf{ofs}} = \phi_1$ cases, respectively, and mostly they are overlapped except for some particular cases. Global offset cases ($\phi_{\mathsf{ofs}} = 180°$) are excluded from the plot, because the metrics explode at early stage for Scheme N, and because the metrics overlaps with other cases for Scheme P and C.





On the other hand, comparison of the convergence rates between Schemes P and C may be effective. For experiments A1 and A2, the convergence rates of $\|E\|_{L_\infty}$ are mostly similar between the two schemes, while that of $\|E\|_{L_2}$ and $\|E\|_{H_1}$ show relatively larger difference. In terms of convergence rates, Scheme P shows slightly larger convergence rates than Scheme C. Therefore, it might be recommended to adopt Scheme P than C, although the difference is still minor. This advantage

should reflect the difference in the formulation of the flux approximation. Scheme P introduces Eq. (29) for computation of the reference longitudes that includes the effect of variation in $\cos\theta$ within a source cell, which is not included in Scheme C.

It is strange that Experiment A3 shows almost the same convergence rates between Schemes P and C. The input field of Experiment A3 has a relatively high-frequency than that of A2, thus the influence of different gradient treatment between the two schemes might show larger impact on the convergence rates. It reflects the characteristics of the input field, where a higher-

frequency wave concentrates in the mid-latitudes and the other area is mostly flat (Ullrich et al., 2009), thus the influence of the gradient is limited.

### 3.4 Additional remarks — applications in past studies

Two reports relating to the second-order conservative remapping scheme based on J99 are worth noting.

Ullrich et al. (2009) present a remapping scheme called Geometrically Exact Conservative Remapping (GECoRe) and show

its performance in idealized cases, comparing it with other schemes, including SCRIP. They find that the error measures in GECoRe and SCRIP deviate significantly for the second-order methods, where the former produces results one or two orders of magnitude better than the latter.

Valcke et al. (2022) present yet another intercomparison study using four remapping algorithms, including SCRIP. A few results obtained by SCRIP are analyzed in the paper, which shows no significant deviation from the other three algorithms, at

least for the second-order conservative remapping. The result plots show that the misfit by SCRIP is the largest among the four algorithms for one benchmark, while it is far less for the other benchmark.

As discussed and demonstrated above, the inconsistent formulation of the reference longitude ($\phi_n$ in Eq. 18) has little impact on the remapping as far as the offset longitude is not far from the midpoint of source cells. It is actually introduced for the other objective, i.e., to avoid multiple-valued longitude at computing difference in longitudes, however it really works as an

side effect to minimize the inconsistency.

It is possible that the inconsistencies relating to offset longitude formulation has some impact on the past studies, however, the author doubts that they explain the behaviors in the above two studies, because the impact on the results is insignificant even when the extreme offset longitude is specified. Rather, different behaviors compared to the other remapping algorithms, if any, should originate from the other part in the J99 algorithm (e.g., computation of intersection). Since, in principle, the effect

of discrepancies in the offset and pivot longitudes is unexpected because the former is not under control of SCRIP, a more detailed exploration of the source code and data is required in order to determine precisely what is happening.



| Scheme | Offset | $\|E\|_{L_2}$ | $\|E\|_{L_\infty}$ | $\|E\|_{H_1}$ |
|--------|--------|---------------|--------------------|---------------|
| N | mid | 1.416 | 1.219 | 0.157 |
| N | edge | 1.367 | 1.221 | 0.099 |
| P | mid | 1.416 | 1.219 | 0.157 |
| P | edge | 1.416 | 1.219 | 0.157 |
| C | mid | 1.409 | 1.221 | 0.131 |
| C | edge | 1.409 | 1.221 | 0.131 |

(a) Experiment A1

| Scheme | Offset | $\|E\|_{L_2}$ | $\|E\|_{L_\infty}$ | $\|E\|_{H_1}$ |
|--------|--------|---------------|--------------------|---------------|
| N | mid | 1.585 | 1.340 | 0.105 |
| N | cell | 1.509 | 1.317 | 0.012 |
| P | mid | 1.585 | 1.340 | 0.105 |
| P | cell | 1.585 | 1.340 | 0.105 |
| C | mid | 1.567 | 1.341 | 0.058 |
| C | cell | 1.567 | 1.341 | 0.058 |

(b) Experiment A2

| Scheme | Offset | $\|E\|_{L_2}$ | $\|E\|_{L_\infty}$ | $\|E\|_{H_1}$ |
|--------|--------|---------------|--------------------|---------------|
| N | mid | 1.782 | 1.628 | 0.827 |
| N | cell | 1.776 | 1.623 | 0.823 |
| P | mid | 1.782 | 1.628 | 0.827 |
| P | cell | 1.782 | 1.628 | 0.827 |
| C | mid | 1.783 | 1.627 | 0.827 |
| C | cell | 1.783 | 1.627 | 0.827 |

(c) Experiment A3

**Table 5.** Convergence rates of three metrics using Schemes N, P, C for experiment (a) A1 (b) A2 (c) A3.



## 4 Summary and conclusion

In this paper, the second-order conservative remapping method on spherical coordinates proposed by J99 is reformulated in an effort to remove the inconsistencies discovered in the original formulation. Two proposals are presented for the valid formulation of the source flux approximation and centroid (or pivot) constraints used to compute the remapping weights. The resulting weights were confirmed to be insensitive to the choice of longitude origin. Until now, the native implementation package of the original algorithm SCRIP has served to mask the inconsistency in the original formulation, as an adjustment to the relative longitude in the SCRIP code has tended to minimize or even erase the problem. Given the adjustment in SCRIP, the author believes that in most practical cases, those using the second-order remapping algorithm in J99 will experience no significant negative impact from the inconsistency problem, especially for cases involving RLL rectangular grid cells. However, it may be prudent for those conducting studies that involve irregularly shaped grid cells or non-modest variable fields and require a high degree of accuracy to review relevant prior studies.

The present study is by no means meant to denigrate past research. To the contrary, the author truly appreciates the contributions of past studies and the accompanying programming packages, which have played an invaluable role in the efforts of the entire climate modeling community. This paper is not intended to discourage but rather to support the validity of past application. If this were not the case, the author would have sought only to develop a new programming package without suggesting revisions to the native SCRIP package. SCRIP-p, a fork of SCRIP, can serve as a drop-in replacement for the original version, acting as a bridge until an official package revision. It should be recognized, however, that SCRIP-p was examined on a somewhat limited basis and for only a few cases. Although it may not fully resolve the fundamental problem for general cases, it is hoped that it will work well as a first trial.

*Code and data availability.* The official package of SCRIP version 1.5 is available from github: https://github.com/SCRIP-Project/SCRIP (last access: 1 April 2024), under an open-source license, with copyright owned by the Regents of the University of California. Details of the license are described in a document of the package. SCRIP-p, a fork of SCRIP, is available from github: https://github.com/saitofuyuki/scrip-p (last access: 1 April 2024), with the same license as the official package, except for where modified, whose copyright is owned by Japan Agency for Marine-Earth Science and Technology (JAMSTEC) under Apache license version 2.0. The exact version of the official and the fork packages, as well as input data and scripts used to produce the results used in this paper, are archived on Zenodo under https://doi.org/10.5281/zenodo.10892796 (Saito, 2024).

*Author contributions.* FS did all the work in the present paper.

*Competing interests.* The author declares that they have no conflicts of interest



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
