# Peer review of "Centroids in second-order conservative remapping schemes on spherical coordinates"

_EGUsphere, 2025_

## Author Comment (AC1)

**Response to the Reviewer comments (RC1)**

I would like to thank Dr. Phil Jones, the reviewer who provided precise and valuable feedback on the manuscript for this resubmission. I addressed all of the points in the responses, and I will submit a revised manuscript that reflects these changes. These changes will significantly improve the quality of the manuscript.

The reviewer comments are quoted in italic with some minor editorial adjustments, followed by responses by the author.

*This paper is a re-submission of an earlier paper that describes a potential problem in the SCRIP high-order conservative remapping algorithm that requires a correction. This revision has a slightly better understanding of the underlying algorithm but still has some significant problems and the testing appears to be incorrect.*

Thank you very much for pointing out many weak points in the manuscript. I will follow your suggestion for the resubmission.

*As in the first submission, the author attempts to derive a flux distribution from a Taylor series in section 2.1. As I wrote in the first review, this derivation is incorrect and the author should not attempt to derive this from a Taylor series. In particular, the constraints in equations 4,5 do not follow uniquely from 3 without additional assumptions. For example, it cannot always be true that a flux evaluated at $c_n$ will be the mean. DK87 actually makes this point in the sentence referenced by the author in which DK87 says it is a Taylor series "only if the mean is assumed to be located at the centroid". Instead, both DK87 and J99 use the distribution of the flux as [can't seem to upload an equation image but latex form is: $f_n = \overline{f}_n + \nabla_n f \cdot (\vec{r} - \vec{r}_n)$ ] which is a construction that automatically satisfies Eq 1 as long as $r_n$ is the centroid so that the second term integrates to zero. By making the leap from his eq 3 to the constraints in 4,5, the author is essentially making the same assumption that the original DK87 and J99 approaches have done by construction, so it's better to start with*

*that anyway and state it as a common constructions that satisfies eq 1. The only reason a Taylor-like expansion is used is so that DK87 can claim the distribution is second-order as long as the gradient is first-order. The author should simply present the distribution as is done in DK87 and J99 to make the assumption explicit and avoid the incorrect derivation in 2.1.*

Now it is really clear, and I fully agree with your suggestion. As you mentioned, I implicitly assumed the relationship between the mean and the centroid point. This should have been mentioned earlier to avoid the leap in the derivation.

*Section 2.2, line 132, the author calls $\phi_c$, $\theta_c$ a reference point and mentions that J99 calls it a centroid. In fact, this is still required to be the centroid and it is not an arbitrary reference point.*

All right. I will rewrite them.

*The actual demonstration of the issue with longitudinal weights appears correct and the pivot fix seems reasonable as shown in Fig 1.*

Thanks! This is the main point of the manuscript.

*An additional proposed fix around line 250 should be removed and the C Scheme later also removed. There is a reason why metric terms are included in coordinate system transformations and relevant operators. Removing the cos(lat) weighting in this fix is likely to introduce significant error in more general meshes, especially near the polar singularities.*

All right, I will remove them from the derivation.

*The biggest problem with the current paper is in the testing section. A couple things are simply oversights: The RLL mesh description in line 337 is wrong and needs to be corrected. Also, they refer to a Scheme O, which I assume was the original formulation which they have now named Scheme N in the previous section. The author needs to pick a consistent name across the sections.*

I was not careful enough to detect the error in the RLL description. I will correct it. Also, I will fixed them to be consistent throughout the paper.

*More seriously, most of the tests use a high-res mesh as the source and perform remaps to coarser meshes. The high-order terms only impact a coarse-to-fine remap so these tests need to use the coarse meshes as the source mesh and the finer mesh as a destination. The fine-to-coarse operation in the tests presented are an averaging and remove most of the impact of the high-order terms. Indeed, in many cases where the overlap is complete, the high-order terms should integrate exactly to zero.*

I really agree with this. Actually, my first manuscript presents the demonstration you suggested, coarse-to-fine remapping. The reason I changed the demonstration to the opposite direction is that one of the reviewers of the first manuscript required it. I will change the demonstration back to the coarse-to-fine case.

*The global offset case does not fix the multi-value longitude issue. It only shifts the problem to a different longitudinal branch cut. So the high errors in these tests are more likely an incorrect correction of the multi-value longitude. If a 2-d map of the solution was shown, I would guess all of the errors would be along a branch cut in the domain and are separate from the error in the formulation that this paper is trying to address. In general, it would be a good idea to show a 2-d greyscale or colormap of the resulting field after a remap in addition to showing the global error norms to demonstrate there are no such artifacts. It's possible they would still not show up in these cases due to the fine-to-coarse averaging, but they would absolutely show up in a coarse-to-fine remap.*

I agree that the global offsetting does not fix the issue for general grids with multi-value longitude. The introduction of the global offset in the manuscript is much limited to the simple configuration presented in the paper. I will explicitly mention this limitation and discuss the issue.

***In the end, there are many issues with SCRIP, including probably the modification in this paper, but this paper requires significant revision to make that case.***

Again, thanks a lot for your precise feedbacks. I will include all your vulnerable suggestions.

---

## Author Comment (AC2)

**Response to the Reviewer comments (RC2)**

I would like to the reviewer 2 who provided precise and valuable feedback on the manuscript. I addressed all of the points in the responses, and I will submit a revised manuscript that reflects these changes. These changes will significantly improve the quality of the manuscript.

The reviewer comments are quoted in italic with some minor editorial adjustments, followed by responses by the author.

*This manuscript revisits the analytical foundation of the second-order conservative remapping scheme presented in Jones (1999, J99). The authors identify an inconsistency in J99's longitudinal term and propose a corrected formulation that resolves the issue. This topic is of continued interest, and clarifying the original derivation is potentially helpful for users and developers of legacy tools derived from J99.*

I appreciate the fair summary of this paper.

*At the same time, I believe the manuscript would benefit from substantial revision to improve clarity of motivation, definitions, and scope. Many statements are broadly phrased and may give readers the impression of general geometric validity, even though the proposed corrections remain tied to the coordinate-specific structure of regular latitude-longitude grids. A clearer presentation would greatly strengthen the manuscript's impact and accessibility.*

Actually, the correction in the numerical scheme I proposed in the manuscript is not limited to regular latitude-longitude cases. To keep things simple, the demonstration in the paper is limited to the RLL case. This may have given readers the impression that the correction has limited application. As you pointed out below, SCRIP computes the weights using line integrals in latitude-longitude space, however, the essential point of the correction in the paper is just before the line integral.

I will improve the manuscript to avoid such misunderstanding.

*I offer the following comments in the hope that they will assist the authors in presenting their valuable observations in the clearest possible*

*way.*

*1. Clarifying the motivation and historical context*
*The manuscript motivates its analysis by stating (Lines 29–30) that J99 "extends the DK87 theory to spherical coordinates, offering an approach that can be applied to any type of grid on a sphere." This wording may lead some readers to infer that J99 handles arbitrary spherical geometry correctly, whereas in practice:*

- *SCRIP, which implements J99, computes weights using line integrals in latitude–longitude space,*
- *and treats edges as straight lines in the coordinate chart,*
- *rather than as great-circle arcs or geodesic boundaries on the sphere.*

*Thus, J99 is applicable to any grid in the sense of an algorithm that can be run on arbitrary inputs, but it does not necessarily handle geometric properties of arbitrary spherical grids appropriately. A more precise description of this distinction may strengthen the manuscript.*

> I will incorporate some of your suggestions in the introduction to SCRIP. However, I am afraid that such notation may be outside the scope of the paper because the main point of the paper is independent of the method used to compute the line integrals. I will strike a good balance.

*Similarly, the discussion of Climate Data Operators (CDO) on Lines 34–40 might be reconsidered. The phrase "have once included" is somewhat ambiguous, and current CDO implementations no longer rely on the SCRIP-based second-order scheme. Today, CDO's first-order conservative remapping is based entirely on YAC. A second-order scheme was also planned, again using YAC, but has not been deployed so far because the available YAC implementation is parallel (MPI-based), and porting it into CDO's primarily serial remapping framework has turned out to be technically nontrivial. This decision was driven by software-engineering constraints, not by any concerns about YAC's geometric formulation. Thus, citing CDO as an example*

*of active reliance on the J99 formulation may be less compelling than intended.*

All right, thanks a lot for your information. I will rewrite the CDO part.

*In addition, more recent work such as the GECoRe paper provides a geometrically exact remapping framework between regular latitude–longitude and cubed-sphere grids, together with closed-form expressions for the relevant integrals (e.g. its Appendix A). It would be helpful if the authors could clearly position the present manuscript relative to such geometry-exact approaches — for example, by explaining that their contribution is a correction within the J99/SCRIP analytical framework, rather than an alternative to fully geometric methods like GECoRe, and by briefly discussing what specific gap or practical advantage this correction offers in that broader context.*

*Nevertheless, because J99/SCRIP remain historically important and are still used in legacy workflows, there is value in clarifying the original analytical inconsistency. Presenting this as a pedagogically motivated correction, and clearly situating it with respect to later developments such as GECoRe and geometry-aware couplers (YAC, TempestRemap, etc.), may resonate better with modern readers.*

Good point. As you mentioned, J99/SCRIP is still widely used. Such a presentation would improve the paper. It won't be easy, but I will include such a discussion.

*2. Clarifying terminology and coordinate dependence*
*(a) Use of "spherical coordinate system"*
*At Line 31, the manuscript refers to "the spherical coordinate system," but the meaning of this term is not formally defined. In practice:*

- *There is no unique "spherical coordinate system"; multiple conventions exist.*
- *Modern conservative remapping frameworks (e.g., YAC, TempestRemap) deliberately avoid relying on latitude–longitude formulas and in-*

*stead express all geometry in 3D Cartesian coordinates, ensuring rotational invariance.*

*The manuscript would be stronger if it were clarified early on that the derivation is performed entirely within the latitude–longitude coordinate chart, and that this choice is intentional.*

> I agree that the definition should be inserted earlier. I will follow your suggestion in the revision.

*(b) Inner product discussion (Lines 112–113)*
*The sentence*
*"The inner product on spherical coordinates is not simply a componentwise product as in Cartesian coordinates..."*
*may inadvertently suggest a misuse that most current geometry-aware remapping codes already avoid. Since $\theta$ and $\phi$ are angular parameters rather than Cartesian axes, they are never combined using componentwise Cartesian logic in modern implementations.*
*If the intention is to emphasize that the gradient operator in spherical coordinates includes metric factors and basis-vector rotation, it may help to express this more precisely. Equation (8) can in fact be written cleanly in terms of 3D vectors, making all inner products well-defined and coordinate-invariant.*

> Yes, the intention is the same as you mentioned. I will rewrite this part, or even remove this part.

*(c) Misuse of the term "centroid"*
*Throughout the manuscript — including in the title — the term centroid is used without a consistent definition. Equation (7) defines the geometric mass centroid of a cell, which lies inside the sphere and might then need to be projected outward onto the spherical surface. However, none of the three schemes introduced later in the paper (Scheme N, Scheme P, or Scheme C) actually computes this geometric centroid (the interior point). Both the "pivot" point $(\theta_p, \phi_p)$ and the "centroid" point $(\theta_c, \phi_c)$ are coordinate-based constructions rather*

*than true geometric centroids on the sphere. Referring to these quantities as centroids is therefore potentially misleading for readers who interpret centroid in the strict geometric sense (i.e. $\mathbf{c}_{\mathrm{geo}} = \frac{1}{A} \int_A \mathbf{x}, dA$, which again lies inside the sphere).*

> This was the most difficult part for me when writing the paper. As you mentioned, the term "centroid" is not clearly defined in the past papers. That is why I introduced a new symbol, the pivot, to avoid using the term "centroid," which could mislead the reader into thinking of the true geometric centroid. The reference point (pivot) introduced in the paper is not directly related to the true centroid inside the sphere; rather, it is a reference point on the surface of the sphere.
>
> I will insert a description of the term "centroid" in the strict geometric sense and try to avoid such misinterpretations.
>
> Also, I will reconsider the title of the manuscript and delete the word "centroid," which is confusing.

*The authors might either need to justify why the flux formulation would remain valid if the reference point were taken to be the geometric centroid (which lies inside the sphere), or explicitly explain that this interior centroid must first be projected back to the spherical surface (e.g. by normalization) if a surface reference point is required.*

> At least for the "pivot" formulation neither is true, because it is derived on the surface. This is really confusing and should be described clearly. I will rewrite the derivation.

*(d) Closed-form expressions and relation to GECoRe*
*It may help readers if explicit closed-form formulas for the reference points in Schemes N, P, and C are included, especially since the manuscript's title centers on "centroids."*
*In this context, it may also be useful to refer to the GECoRe paper, which provides closed-form expressions (and derivations) in its Appendix A. The GECoRe paper also presents a geometrically exact regridding solution between regular latitude–longitude and cubed-sphere*

*grids. It would be beneficial to include a brief comparison between the present manuscript and the GECoRe approach — for example, by pointing out the potential gap between the two methods and discussing the potential benefits of the present work relative to GECoRe.*

First, the term 'centroid' will be probably removed from the title in the revision, thus, the close-form formulas may be much beyond the scope of the paper. However, the comparison between GECoRe approach and the present paper might be interesting. I will include this matter.

*3. Scope and generality*
*Two limitations of the current formulation deserve clearer acknowledgment:*
*(a) The corrections apply only within the coordinate framework of J99/SCRIP*
*Because SCRIP uses a latitude–longitude planar approximation, the corrected formulas presented here remain applicable only to the same class of structured grids. The method does not extend to arbitrary, non-uniform unstructured grids. A brief comment on this limitation may prevent misinterpretation.*
*(b) The proposed centroid/pivot definitions are meaningful only for regular RLL grids*
*Area-weighted averages of $\theta$ and $\phi$ (even with a $\cos\theta$ weight for longitude) do not correspond to geometric centers for general spherical polygons. Their interpretation becomes ambiguous when cells are skewed, large, or irregular.*
*These points do not invalidate the proposed correction — which is internally consistent within the J99 framework — but clarifying scope will improve the precision of the manuscript.*

(a) Partially, yes. The correction in the present paper may only be applicable to the same class of structured grids as you pointed. However, as I mentioned earlier, the correction maintains the consistency of the remapping scheme, regardless of the grid structure. (b) Partially yes. I agree that for

many irregular cells, area-weight mapping cannot be a proper approximation. However, there are regular grids that are not LL-shaped which can be the target of the corrected scheme. This is a really important point, and I will improve the description in the revision to make the limitations of the correction clearer.

*Minor Comments*
*Line 48: The phrase "serious damage to the remapping result" would benefit from a concrete explanation or example illustrating the numerical consequences.*

The first submission includes such illustration. I will reconsider to include it again.

*Some expressions, such as those involving coordinate shifts ($\phi_{ofs}$), could be clarified to ensure readers understand exactly how invariance is being tested.*

I will rewrite the part to make it clearer.

*The manuscript might benefit from reorganizing Sections 2.1–2.2 to distinguish clearly between the flat Cartesian case and the spherical-coordinate case.*

I will reorganize the sections as you suggested.